# A First Insight into the Microbial and Viral Communities of Comau Fjord—A Unique Human-Impacted Ecosystem in Patagonia (42° S)

**DOI:** 10.3390/microorganisms11040904

**Published:** 2023-03-30

**Authors:** Sergio Guajardo-Leiva, Katterinne N. Mendez, Claudio Meneses, Beatriz Díez, Eduardo Castro-Nallar

**Affiliations:** 1Departamento de Microbiología, Facultad de Ciencias de la Salud, Campus Talca, Universidad de Talca, Avda. Lircay s/n, Talca 3465548, Chile; 2Centro de Ecología Integrativa, Campus Talca, Universidad de Talca, Avda. Lircay s/n, Talca 3465548, Chile; 3Center for Bioinformatics and Integrative Biology, Facultad de Ciencias de la Vida, Universidad Andrés Bello, Santiago 8370186, Chile; 4Departamento de Genética Molecular y Microbiología, Facultad de Ciencias Biológicas, Pontificia Universidad Católica de Chile, Santiago 8331150, Chile; 5Departamento de Fruticultura y Enología, Facultad de Agronomía e Ingeniería Forestal, Pontificia Universidad Católica de Chile, Santiago 8331150, Chile; 6ANID—Millennium Science Initiative Program—Millennium Nucleus for the Development of Super Adaptable Plants (MN-SAP), Santiago 8370186, Chile; 7Center for Climate and Resilience Research (CR)2, Santiago 8370449, Chile; 8Millennium Institute Center for Genome Regulation (CGR), Santiago 7800003, Chile

**Keywords:** estuarine waters, coastal microbiome, Patagonia, shotgun metagenomics

## Abstract

While progress has been made in surveying the oceans to understand microbial and viral communities, the coastal ocean and, specifically, estuarine waters, where the effects of anthropogenic activity are greatest, remain partially understudied. The coastal waters of Northern Patagonia are of interest since this region experiences high-density salmon farming as well as other disturbances such as maritime transport of humans and cargo. Here, we hypothesized that viral and microbial communities from the Comau Fjord would be distinct from those collected in global surveys yet would have the distinctive features of microbes from coastal and temperate regions. We further hypothesized that microbial communities will be functionally enriched in antibiotic resistance genes (ARGs) in general and in those related to salmon farming in particular. Here, the analysis of metagenomes and viromes obtained for three surface water sites showed that the structure of the microbial communities was distinct in comparison to global surveys such as the Tara Ocean, though their composition converges with that of cosmopolitan marine microbes belonging to Proteobacteria, Bacteroidetes, and Actinobacteria. Similarly, viral communities were also divergent in structure and composition but matched known viral members from North America and the southern oceans. Microbial communities were functionally enriched in ARGs dominated by beta-lactams and tetracyclines, bacitracin, and the group macrolide–lincosamide–streptogramin (MLS) but were not different from other communities from the South Atlantic, South Pacific, and Southern Oceans. Similarly, viral communities were characterized by exhibiting protein clusters similar to those described globally (Tara Oceans Virome); however, Comau Fjord viromes displayed up to 50% uniqueness in their protein content. Altogether, our results indicate that microbial and viral communities from the Comau Fjord are a reservoir of untapped diversity and that, given the increasing anthropogenic impacts in the region, they warrant further study, specifically regarding resilience and resistance against antimicrobials and hydrocarbons.

## 1. Introduction

Coastal and estuarine environments are a hotspot for disturbances of anthropogenic origin [1,2,3]. Researchers worldwide have reported pollution by hydrocarbons, antibiotics, and other compounds (organochlorine pesticides and heavy metals) in bodies of water near cities and ports. However, relevant concentrations of pollutants have also been reported in places traditionally regarded as pristine [4,5,6,7,8,9], such as Patagonia. With nearly 40% of the population living within 100 km of a coastline, coastal and estuarine environments are a boundary between terrestrial and aquatic (freshwater and marine) ecosystems [10]. These environments bridge human health and the health of the environment, for instance, through emerging pollutants such as antibiotics and antimicrobial resistance genes (ARGs) [1,11]. In particular, ARGs might cycle over environmental and human hosts, since many ARGs are harbored in mobile genetic elements that are mobilized through horizontal gene transfer [11,12,13,14].

There is a global interest in monitoring ARGs in the natural and built environment [15,16,17]. Recent studies reveal the global prevalence of ARGs in natural systems such as mangroves [18], rivers [19], and other water bodies, and some link their prevalence to fecal as well as industrial pollution [20,21]. In the built environment, global surveys have characterized the microbiome and resistome (collection of all ARGs) in the urban environment and transport systems [15]. Farms and other agriculture facilities, as well as food processing plants, are also a focus of the study of ARG prevalence, type, and abundance [22,23,24]. As with naturally occurring resistomes, ARGs are often found to be prevalent but not highly abundant [15,25]. 

While ARGs are naturally occurring and ancient, e.g., presence of vancomycin resistance genes in 30,000-year-old permafrost in the Yukon, humans have contributed to their current ubiquity and prevalence [26,27]. The consequences of the widespread nature of ARGs are two-fold. First, ARGs presence, prevalence, and abundance serve as indicators of human activity [28,29,30]. Secondly, natural and built environments might serve as ARG reservoirs whose genes may or may not be horizontally transferred to human pathogens that are currently susceptible to antibiotic treatment [31]. This is especially relevant in industries that use antibiotics from the clinical classes. In Patagonia, for instance, salmon farming uses large quantities of antibiotics such as oxytetracycline and florfenicol on fish in open cages in fjords and channels, both compounds with close chemical analogs in human medicine [32,33,34,35].

On the other hand, in coastal and estuarine environments in Patagonia and other relatively pristine environments, microbial and viral biodiversity is poorly understood, especially in comparison to global open ocean surveys. Expeditions such as Tara Oceans, Malaspina, the Global Ocean Sampling, and GEOTRACES have generated terabytes of shotgun metagenomic information, uncovering immense microbial and viral diversity as well as environmental drivers, metabolic strategies, and functional diversity [36,37,38,39,40]. We have spatially and temporally explicit microbial community data covering all of the world’s oceans. However, continental coastal areas alone amount to more than 700,000 km and remain underexplored regarding microbial community diversity and function, in particular in pristine areas [41].

Patagonia, i.e., the region of southern South America from 40° S and higher, is traditionally regarded as pristine, even though industries such as agriculture (livestock) and salmon farming are commonplace. Similarly, due to the fragmented nature of the geography, land access to Patagonia is hampered by fjords and channels, especially west of the Andes Mountains. This results in commercial and passenger transport conducted mainly by sea routes on cruise ships, ferries, and small boats, all of which use hydrocarbons as a power source.

Northern Patagonia, including the Comau Fjord, is impacted by high-density salmon farming, which results in the export of particulate organic matter to the sediments and an increase in dissolved inorganic nitrogen and dissolved organic nitrogen, all of which have been associated with harmful algal blooms. Additionally, the salmon farming industry poured more than 370 tons of antibiotics into the ocean in 2020, which is a hazard to local biodiversity and possibly to human health [42].

In this study, we hypothesized that the Comau Fjord viral and microbial communities will be distinct from those collected in global surveys, but they will bear the hallmarks of microbes from coastal and temperate regions. Particularly to this study, we also hypothesized that microbial communities will be functionally enriched in ARGs in general and in those related to salmon farming in particular. For this, we sampled three surface coastal sites in the Comau Fjord and obtained metagenomes and viromes to test whether this under-sampled part of the world offers any taxonomic and functional diversity as compared to a global survey of the open ocean, in particular those publicly available from the Tara Oceans expedition. We also compared our results against global databases such as IMG, NCBI’s RefSeq, and GOS to obtain taxonomic and biogeographical information and functionally analyzed the metagenomes regarding functions and their resistomes, as well as viral communities, which showed a certain degree of uniqueness in their protein content.

## 2. Materials and Methods

### 2.1. Sample Collection and Sequencing

We collected 40 L of water at 5 m below sea level at 3 sampling sites (each 3 km apart; from a boat) at the head of the Comau fjord located in Northern Patagonia (42° S) using an electric water pump (Pedrollo, San Bonifacio, Italy). The fjord is characterized by seasonal variability associated with freshwater inputs from nearby rivers (the Blanco, Cisnes, Lloncochaigue, and Vodudahue rivers), a mean depth of 250 m (a maximum of ~480 m), 41 km of length and 4.5 km of width, precipitation (>5000 mm), and ice thawing in the summer season. During water collection in February 2016, we used a 50 μm mesh to avoid macroscopic debris. Water samples were immediately stored at 4 °C for no more than 48 h until processing. Then, we assembled a filtration system using a peristaltic pump to run the 40 L of water from each sample through a 0.22 μm Sterivex filter to capture prokaryotic organisms and subsequently through a 100 kDa tangential concentrator to concentrate viral particles. These resulted in prokaryotic and viral fractions for each of the 3 water samples, named Bac S1/S2/S3 and Vir S1/S2/S3, respectively. 

We used phenol-chloroform to isolate DNA from both prokaryotic and viral fractions. In addition, prokaryotic DNA was obtained by Xanthogenate nucleic acid isolation, as described in Alarcón-Schumacher et al., 2019. Viral DNA was extracted from viral particles purified by a CsCl gradient as described in Guajardo-Leiva et al., 2021 [43,44]. DNA integrity was checked by capillary electrophoresis using a Fragment Analyzer^TM^ System (Agilent Technologies, Santa Clara, CA, USA). DNA was quantified in a Qubit fluorometer and used for the preparation of paired-end libraries according to the Illumina sample preparation guide. Libraries were subsequently sequenced on an Illumina MiSeq instrument at the Plant Biotechnology Center, Universidad Andrés Bello (Santiago, Chile). Subsequently, we evaluated the quality of the 250 bp long paired-end reads using FastQC v0.11.8 and performed the filtering and trimming of the raw data using PRINSEQ v.0.20.4 (*-min_qual_mean 20 -ns_max_p 0 -lc_method dust -lc_threshold 30 -trim_qual_left 20 -trim_qual_right 20 -derep 1*; quality score threshold > 20) [45,46]. We used the quality-controlled sequences for the following analyses.

### 2.2. Water Nutrients, Elemental Composition, and Physicochemical Analyses

Seawater was collected by pumping through a 50 μm net to exclude large organisms and stored in dark pet drums until arrival at the laboratory. Two liters of seawater were filtered through two 0.7 μm precombusted GF/F filters (Merck-Millipore, Burlington, MA, USA) and stored in the dark at −80 °C until chlorophyll-a (Chl-a) extraction and organic carbon and nitrogen determination (POC and PON). Additionally, 15 mL (in triplicates) of seawater was filtered through 0.2 µm mixed cellulose filters (Sartorious) and collected in polyethylene bottles for salinity, nitrate, nitrite, phosphate, and silicic acid determination. Bottles were stored at −20 °C until analyses.

Total Chl-a was quantified by methanol extraction and spectrophotometric determination. Each filter was incubated in 1.4 mL of hot methanol (65 °C) and bead-beaten with 1 mm glass beads (2 pulses of 1 min at 4000 rpm). The mix was incubated in the dark for 5 min at 65 °C and 15 min on ice. After the incubation, tubes were centrifuged at 15,000× *g* for 15 min to remove sediment filter fragments. One milliliter of the supernatant was spectrophotometrically measured at 665 and 750 nm. To determinat the Chl-a content, we used the equation described by Marker et al., 1980, with a constant value of 12.99 [47]. POC and PON were analyzed by isotope ratio mass spectrometry (Thermo, Delta Advantage IRMS, Waltham, MA, USA) coupled to an organic elemental analyzer (Thermo, Elemental Flash EA2000) in the Laboratory of Biogeochemistry and Applied Stable Isotopes (LABASI), Pontificia Universidad Católica de Chile, Santiago, Chile. Nitrate, nitrite, phosphate, and silicate concentrations were measured automatically by a Technicon AutoAnalyzer^®^ (AA3 Seal Analytical, North Queensland, Australia), and salinity by a refractometer (Argent) in the Laboratory of Biogeochemistry of Greenhouse Gases (LABGEI) of the Pontificia Universidad Católica de Valparaíso, Valparaiso, Chile. 

Variables such as temperature, light intensity, and pH were measured in situ at the sampling sites. Temperature and light were recorded every five minutes using a data logger, the HOBO Pendant^®^ (Onset Computer Corporation, Bourne, MA, USA), and pH was measured using a pH indicator, Hydrion (9800), and a Spectral 0–14 Plastic pH Strip (Micro Essential Laboratory, New York, NY, USA).

### 2.3. Distance-Based Analysis and Comparison with Global Ocean Data

In order to compare our metagenomes with global ocean data, we downloaded all of Tara Ocean’s surface samples (245 bacterial metagenomes and 112 viromes; http://www.taraoceans-dataportal.org) and performed a distance-based analysis. We used MASH v1.1.1 (*mash sketch -m 2 -k 21; mash dist*), which uses the MinHash dimensionality-reduction technique to cluster the metagenomic sequences and calculates the Jaccard Index to report an all-against-all distance matrix for all metagenomic samples [48].

### 2.4. Taxonomic Composition Metagenomes and Viromes 

We performed a read-based taxonomic analysis for both metagenomes and viromes using the PathoScope v2.0 pathomap module. PathoScope uses Bowtie v2.2.9 for the read-mapping step and the PathoScope v2.0 pathoid module for the estimation of read counts using a Bayesian model for read reassignment [49,50]. For the metagenomes, we aligned the reads against the prokaryotic RefSeq representative genome database and the 2631 draft metagenome-assembled genomes from the Tara Oceans [51]. For the viromes, we aligned the reads against all viral RefSeq genomes (8957 viral genomes) [52], the 125,842 viral contigs from the JGI’s Earth Virome study, release IMG_VR_2018-07-01_4 (IMG/VR) [53], and the 298,383 viral contigs from the Global Oceans Viromes (298,383 viral contigs) (GOV) databases [54]. For both metagenomes and viromes, we used the phix174, hg19 (human), chloroplast, mitochondrion, and fungi RefSeq genomes as filter databases at the read-mapping step in order to discard possible contaminating sequences.

### 2.5. Metagenomes Functional Potential and Presence of AR Genes

For the functional annotation of metagenomes, we aligned all quality-controlled reads against the SEED subsystem database using DIAMOND v0.9.21, as implemented in SUPER-FOCUS v0.31 (available through bioconda repositories; miniconda3 v4.3.27.1), and reported read abundance for each subsystem level 1, 2, 3, and function [55,56,57]. To estimate the presence and abundance of antibiotic resistance (AR) genes, we used the ARG_OAP pipeline v2.0, which uses BLAST v.2.7.1 to align the reads against the SARG database and estimate the relative abundance of AR gene sequences expressed as the number of reads mapping against a certain antibiotic resistance gene per million of total reads [58,59]. The SARG database was built by the ARG_OAP developers, and it includes the CARD (Comprehensive Antibiotic Resistance Database) and ARDB (Antibiotic Resistance Genes Database) [60,61], as well as selected AR gene sequences from the NCBI-NR database.

### 2.6. Viromes Protein Clusters

We assembled the quality-controlled reads from each viral sample (Vir S1/S2/S3) using SPAdes v3.8.0 (*--meta*), then obtained the protein sequences from the viral contigs using Prokka v1.12 *(--kingdom viruses --metagenome*), which uses Prodigal v2.6.3 for the prediction of open reading frames (ORFs) [62,63,64]. We clustered the Patagonia viral proteins > 60 amino acids in length along with the Tara Oceans Viromes (TOV) proteins using CD-HIT v4.6 with the same parameters used for TOV protein clustering (*cd-hit-2d -g 1 -n 4 -d 0*, 60% identity, and 80% coverage) [65]. We classified the Patagonia viral protein clusters present in TOV samples according to the Tara metadata using an in-house script available at https://github.com/Vasco-Varas/Bioload (1 November 2022). Then, we used the Patagonia viral proteins that were not similar to any TOV proteins to form new clusters with two or more ORFs (Patagonia Protein Clusters (PTGN PCs)), using CD-HIT under the same settings as the TOV (*cd-hit -g 1 -n 4 -d 0 -G 0*, 60% identity, and 80% coverage).

### 2.7. Analysis and Visualization

We analyzed and visualized all results obtained through the methodologies described above using the R software along with the following packages: ampvis2, ggfortify, ggplot2, ggpubr, ggrepel, gridExtra, phyloseq, RColorBrewer, reshape2, taxize, tidyverse, and VennDiagram packages [66,67,68,69,70,71,72,73,74,75,76,77].

## 3. Results

In order to compare the diversity of the microbial and viral communities of Patagonia surface waters against marine global samples, we conducted a distance-based analysis using MASH distances. We used the Tara Oceans metagenomes due to the sampling density and cover of global oceans and found that Comau samples were more distinct at the genomic level than those from global oceans (Figure 1). Microbial communities from the Comau Fjord were more closely related to surface samples from the South Atlantic, South Pacific, and Southern Oceans (Figure 1A; Appendix A). Similarly, a comparison against global viromes showed that the Comau Fjord viral communities were also genetically distinct from global communities, though more closely related to Southern Ocean and South Atlantic Ocean samples from surface waters and from the deep chlorophyll maximum (DCM; Figure 1B).

In order to get insights into the members of the microbial and viral communities in the Comau Fjord, we conducted a read-based taxonomic profiling using PathoScope2. Using the RefSeq and Tara databases, we found that Comau samples were dominated by proteobacteria, bacteroidetes, and actinobacteria. In particular, communities were dominated by *Planktomarina temperata*, which reached up to 30–40% relative abundance (Figure 2A). In contrast, the taxonomy of viral communities yielded poor results, with read classification rates of <1%. For this reason, we also classified viral reads using a collection of viral genomes from JGI’s IMG/VR and from the Global Ocean Viromes (GOV) databases annotated with location and biogeographic information (metadata-enabled functional profile). Most of the Comau viral sequences matched viruses from coastal and neritic zones (Figure 2B), with a few members found in open oceans. Most abundant viruses were cosmopolitan, as they have been observed, for instance, in Delaware, Chesapeake Bay, the Subarctic Pacific Ocean, and the North Sea, as well as in the South Pacific and Atlantic Oceans. Overall, we observe that microbial communities exhibit dominant members, while viral communities are more equitable (top abundant taxa ~5%).

Additionally, we sought to determine whether the taxonomically similar microbial communities from the Comau Fjord were convergent or divergent with regard to functional diversity. Comau Fjord microbial communities were similar not only at the taxonomic level but also at the functional level. Around 3% of the read abundance corresponds to genes related to virulence, of which the most abundant category (7.9%; subsystem level 2) is resistance to antibiotics and toxic compounds (Figure 3A). Within this category, ~40% corresponds to functions associated with antibiotic resistance, such as multidrug efflux pumps (19.21%), resistance to fluoroquinolones (15.9%), and beta-lactamases (5.33%). Given that the sampling area is dotted with open-cage aquaculture facilities that use large amounts of antibiotics, we also profiled the resistome of the Comau Fjord microbial communities and compared it for reference against samples from the surface and DCM from the Southern Ocean, South Atlantic, and South Pacific Oceans (Figure 3B). We found that Comau samples exhibited resistance mainly to beta lactams and tetracycline (which shared a molecular mechanism with oxytetracycline), to bacitracin, and to the group macrolide–lincosamide–streptogramin (MLS). We also found that their relative abundance was comparable to other samples not exposed to large quantities of antibiotics (Figure 3B), which suggests that the effect of aquaculture on the resistome is not captured by our samples and might be more local and acute.

Finally, we wanted to functionally profile the viral communities in the Comau Fjord. Environmental viromes, however, are difficult to profile due to intrinsic features of viruses, i.e., high substitution rates, large population sizes, high turnover, as well as practical issues such as a lower representation in databases than human or industry-related viruses. Thus, we decided to use the membership of protein clusters (PCs) (same methodology as in [78,79]) to gain insight into the novelty of PCs from the Comau Fjord, Patagonia (PTGN PCs) in comparison to those available worldwide (Figure 4; Appendix A). Comau Fjord viromes shared 30.4% of identified protein clusters (TOV and PTGN PCs) and exhibited approximately 10% of sample unique clusters (Figure 4A). Out of the 94,418 PCs, only 51,104 TOV PCs (54.1%) were represented in the Comau samples and corresponded with TOV PCs from different biogeographic regions in the deep chlorophyll maximum and surface zones but not in the mesopelagic zone (Figure 4B). Comau proteins not belonging to any TOV PC (311,837) were de novo clustered to form 43,314 novel PCs (termed PTGN PCs), of which 37.8% were shared (the core; Figure 4C). Proteins that did not form any clusters were considered singletons (196,233) (Figure 4C; Appendix A).

## 4. Discussion

In this study, we compared the microbial and viral communities of coastal surface waters in the Comau Fjord with those of the global oceans. We detected markedly different composition and structure in these Patagonian microbial and viral communities. Coastal and estuarine waters differ from the open ocean in their oceanographic characteristics as well as in their physical parameters. In particular, estuarine waters such as those of the Comau Fjord are subject to salinity changes due to glaciers melting and snow and precipitation (>5000 mm in the study site) [80], which might select for and foster microbial communities whose members are able to thrive under fluctuations in osmotic conditions. In addition, natural inputs of nutrients, sediments, and inorganic material are warranted in estuarine environments and are more relevant than in the open ocean. These two variables indeed contribute to the observed differences in community composition and structure; however, studies show that anthropogenic disturbances are key drivers as well. For instance, Chen et al. (2022) showed that microplastics not only alter the microbiome in marine sediments but also have functional consequences, e.g., altering nitrous oxide production (increasing) and associated pathways [81]. Interestingly, some studies have also shown that the effects of anthropogenic disturbances can affect specific groups of microbes. For instance, Jasmin et al., 2020 showed that heavy metals might influence the abundance and occurrence of specific taxa, especially cyanobacteria [82]. In complex estuarine environments, such as those near cities and ports, wastewater is added to estuaries, usually through riverine systems, which results in increased fecal matter bacterial indicators. Orel et al. (2022) showed that bacterial indicators of anthropogenic pollution in coastal ecosystems in the Mediterranean are decoupled in their occurrence and abundance from the natural seasonal variation expected in estuaries [83]. In Northern Patagonia, Olsen et al. (2017) showed using microcosm experiments that the addition of nutrients alone by salmon farming, e.g., NH4 and PO4, affects community composition and structure but not diversity, suggesting that microbial communities in Patagonia are resilient but not resistant to disturbances afforded by salmon farming [84]. Other studies have looked at the effect of salmon farming on sediment microbial communities, where similar effects have been recorded [85], which also holds true for microeukaryotes [86]. Further studies in Patagonia should be temporally and spatially explicit so as to tease apart microbes and viruses, and their genes, coming from anthropogenic activities from naturally occurring and seasonal ones.

We found similar patterns when looking at viral communities in the Comau Fjord. While viral communities were genetically distinct from those from the global oceans, a closer characterization showed that Comau viruses were cosmopolitan and had relatives mainly from the South Atlantic Ocean and Subarctic Pacific Ocean. Interestingly, higher-abundance viruses were taxonomically related to riverine and estuarine viruses from other continents, suggesting that environmental factors shape their evolution in a convergent fashion. For instance, Patagonia viruses were related to viruses described in the Chesapeake Bay, Delaware Coast, Guaymas Basin, Delaware River and Bay, Monterey Bay, and Columbia River. Similar to our findings, Sun et al. (2021) found a large number of diverse and abundant viruses without culturable representatives that were distinct among the Chesapeake Bay and Delaware viral communities and, in turn, highly differentiated from ocean viromes [87]. Similarly, in Antarctica, Alarcón-Schumacher et al. also found that viral communities in the Southern Ocean are highly divergent at the genomic level but constitute taxa with global distribution [88]. Altogether, our results agree with those presented elsewhere and suggest that estuarine microbial and viral communities represent an untapped reservoir of taxonomic novelty distinct from that of the open ocean and vulnerable to anthropogenic disturbances.

In this study, we also addressed the resistome of the microbial communities in the Comau Fjord, especially in relation to salmon farming. Functional profiles showed that Comau microbial communities were enriched in amino acid and protein metabolism genes, as well as other homeostatic functions such as carbohydrate and DNA metabolism, cofactors, vitamins, and others, as well as RNA metabolism (SEED Subsystem Level 1). Interestingly, a high proportion of virulence genes were assigned to resistance to antibiotics (SEED Subsystem Level 2), of which almost 40% belong to multidrug resistance efflux pumps, resistance to fluoroquinolones, and to beta-lactamases. Several studies have shown that salmon farming and aquaculture in general might promote the emergence and spread of antimicrobial resistance in sites far removed. For instance, Sha et al. (2014) analyzed 200 bacterial isolates from aquaculture and non-aquaculture sites in southern Chile and showed high levels of ARGs in both types of sites, of which some were able to transfer their resistance to *Escherichia coli*, highlighting the potential risk to human health [89]. Similarly, the gut microbiome of farmed fish is also shaped by salmon farming. Higuera-Llantén et al. (2018) isolated bacteria from the gut of salmon obtained from fish farms in southern Chile and found high levels of antibiotic resistance to florfenicol and oxytetracycline, demonstrating that animal gut microbiomes are also affected [33] (though at low risk for humans [33]).

We also specifically searched for ARGs in our dataset and compared it to regional samples from the South Atlantic, South Pacific, and Southern Oceans (Tara Ocean survey; samples closely related to Comau samples) to get insights as to the increased abundance or prevalence of ARGs in the Comau Fjord. Although our results were congruent with the literature, i.e., macrolide−lincosamide−streptogramin, multidrug, betalactams, and others [33,89,90,91], resistome profiles of regional samples without the influence of salmon farming showed similar patterns, if not more abundant ARGs. This might be explained due to the seasonal point exposure of microbial communities from the water column to antibiotics in the study site, as well as current dynamics in open ocean samples. Salmon farming facilities use antibiotics in the feed, especially after fish are transferred from freshwater to marine in the span of days (pulse disturbance) rather than continuously (press disturbance) [92]. Recent studies have profiled microbial communities using metagenomics to characterize the diversity and abundance of ARGs in the oceans and other water bodies and found similar patterns. Cuadrat et al., 2020 analyzed the Tara Oceans dataset and found high diversity of ARGs, however, at low relative abundance [93] (see also [94]). In the urban environment, a global survey involving a large dataset collected over time in 60 cities showed that ARGs were highly prevalent but low in relative abundance [15]. We suggest that estuarine waters in Patagonia and the Comau Fjord are sources of ARGs and that the effect of salmon farming on the resistome is local and acute. Studies using experimental or semi-experimental approaches, e.g., mesocosms, might be better suited to shed light on the effect of antibiotics on the microbial communities of the Comau Fjord [94].

Viruses are important yet often neglected members of the ocean microbiome, which in part is derived from the difficulty in classifying their genes and genomes [95]. Here, we build on a framework used by other authors to, instead of searching for high-level taxonomic classifications (many of which are meaningless), focus on metadata-enabled functional inquiries of the great viral unknown [79]. In this study, we addressed the great unknown problem by building a set of Comau Fjord Patagonia protein clusters (PTGN PCs) and determining whether they are shared with other viromes or rather novel. While our samples were taken 3 km apart from each other, the core PTGN PCs accounted for only 37.84% of the total number of PTGN PCs, suggesting that estuarine waters in the Comau Fjord are highly heterogeneous in viral composition and gene content. Recent studies at the global scale have found that ocean viral communities are indeed diverse and heterogeneous, sometimes not fitting ecological models such as the latitudinal diversity gradient [96]. At the local scale, studies focusing on coastal viral communities show high turnover of viruses regardless of host taxa and temporal dynamics, suggesting that viruses are continuously lysing abundant prokaryotes via viral infection [97]. Interestingly, proteins from Comau viromes were found in global PCs (~30%), indicating a large, shared gene pool exists between Comau viromes and global viral communities. Further studies need to address the origin of coastal and estuarine viral communities, and these are undoubtedly seeded in part by rivers and runoff from terrestrial ecosystems nearby. Finally, we obtained a large proportion (~40%) of singleton PCs and ~30 thousand novel PCs (not present in global surveys). These singletons could either be artifacts or more likely reflect members of the rare biosphere, which is unfortunately under sampled in both our own and published studies. 

In this study, we tested whether Comau microbial and viral communities would be distinct in composition and function in comparison to those collected in global surveys; however, we expected they would bear the hallmarks of microbes from coastal and temperate regions. Our results support this hypothesis, as the Comau Fjord microbial and viral communities are diverse and different from those sampled in global ocean surveys. Similarly, most published global ocean surveys are densely sampled in regions other than cold-water and high-latitude oceans, which would partially explain the novelty of the Comau Fjord and locations alike. We also hypothesized that Comau microbial communities will be functionally enriched in ARGs in general and in those related to salmon farming. This was partially confirmed by our data since we were able to find genetic determinants of resistance against florfenicol and oxytetracycline (the two most widely used antibiotics in salmon farming in Chile). However, our study presents limitations. More dense sampling with a spatial and temporal design might be able to better capture any signal of disturbance from salmon farming. In addition, microbes attached to particles as well as sediment samples might yield a higher resolution picture of the anthropogenic disturbance caused by salmon farming. Finally, a meta-analysis of estuarine waters, fjords, and channels with comparable methodologies might provide further insight as to the effect of anthropogenic disturbances on the microbial and viral communities of the water column.

Since most human activities occur in coastal areas, such as bays, sounds, fjords, and estuaries, they represent a unique ecosystem that needs to be further studied to understand the interplay between human and environmental health. In particular, determining whether microbial communities are resistant or resilient to disturbances of anthropogenic origin, i.e., diesel contamination and antibiotics, will allow us to understand to what extent their functions will also be affected in the future.

## Figures and Tables

**Figure 1 microorganisms-11-00904-f001:**
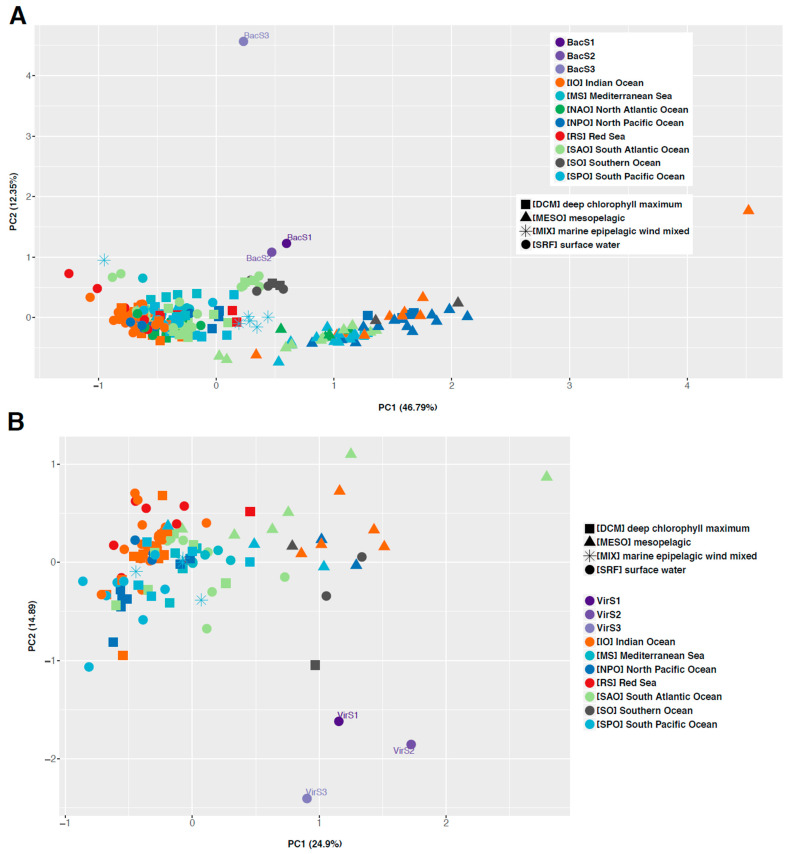
Principal component analysis of metagenome MASH distances from the Tara Oceans dataset and Comau Fjord samples for metagenomes (**A**) and viromes (**B**).

**Figure 2 microorganisms-11-00904-f002:**
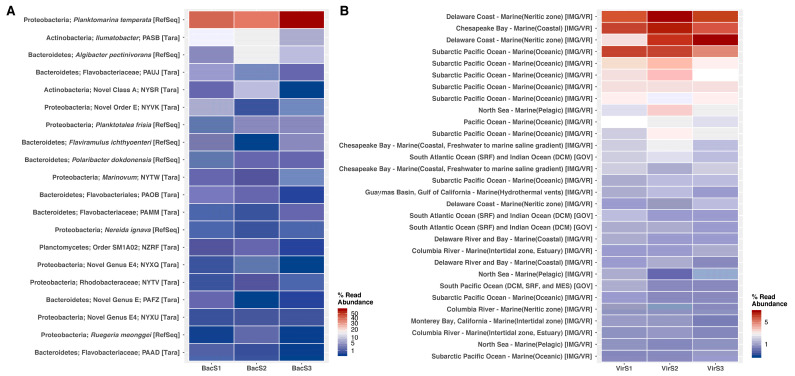
Taxonomic profile of the top 20 most abundant taxa for metagenomes and viromes from the Comau Fjord. Taxonomic information from bacteria was derived from NCBI’s RefSeq database and metagenome-associated genomes from Tully et al., 2018 (**A**). Virome metadata was derived from IMG/VR and GOV databases (**B**).

**Figure 3 microorganisms-11-00904-f003:**
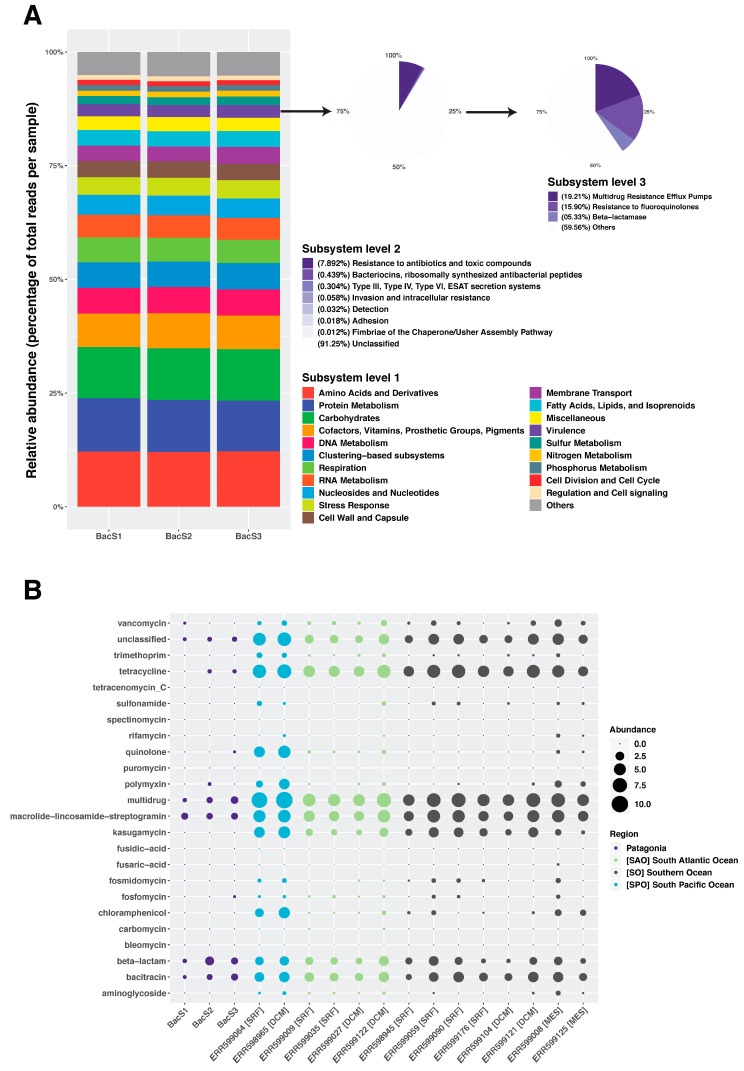
Functional profiling of Comau Fjord microbial communities at the three SEED subsystem levels (**A**) and their antibiotic resistance profiles (**B**). Functions are expressed as relative abundance of mapping reads against the SEED database in (**A**) and as number of gene sequences per million in (**B**).

**Figure 4 microorganisms-11-00904-f004:**
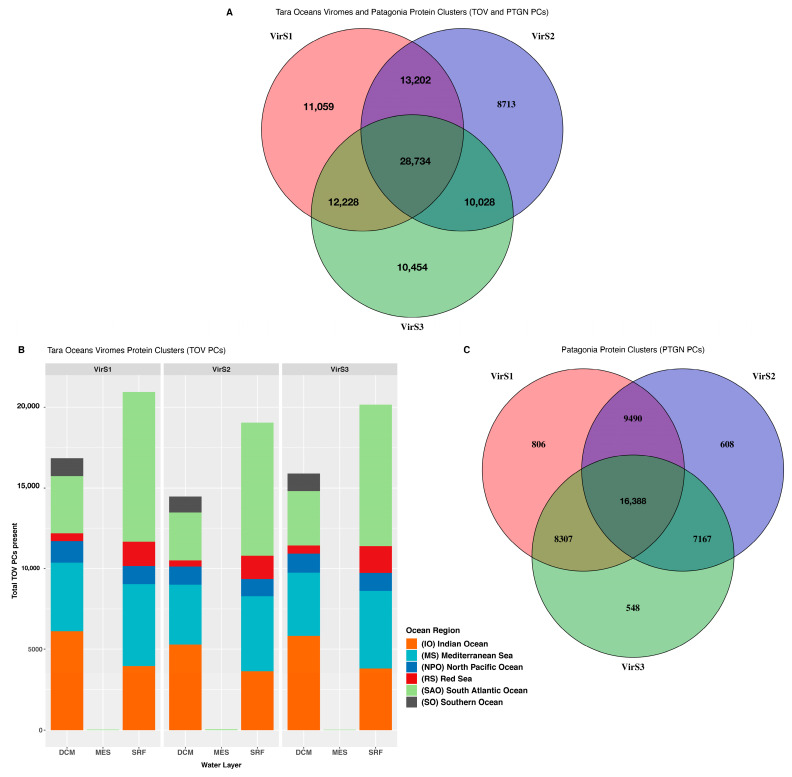
Protein cluster (PC) membership for Comau Fjord virome proteins. Comau Fjord viromes shared 30.4% of identified protein clusters (TOV and PTGN PCs) and exhibited approximately 10% of sample unique clusters (**A**). Out of the 94,418 PCs, only 51,104 TOV PCs (54.1%) were represented in the Comau Fjord samples and corresponded to TOV PCs from different biogeographic regions in the deep chlorophyll maximum and surface zones but not in the mesopelagic zone (**B**). Comau Fjord proteins not belonging to any TOV PC (311,837) were de novo clustered to form 43,314 novel PCs (termed PTGN PCs), of which 37.8% were shared (core; **C**). Proteins that did not form any clusters were considered singletons (196,233) (**C**).

## Data Availability

This whole metagenome shotgun project has been deposited in GenBank under the accession no. PRJNA359936. The version described in this paper is the first version.

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
