# Peer review of "A First Insight into the Microbial and Viral Communities of Comau Fjord—A Unique Human-Impacted Ecosystem in Patagonia (42 S)"

_microorganisms, 2023, doi:10.3390/microorganisms11040904_

Round 1

Reviewer 1 Report

The paper by Guajardo-Leiva regards the anthropogenic impact on the bacterial and viral particle diversity and function in a Patagonian estuary. At first glance the paper seems well written and composed. However even if the methodological approach is sound there is a major flaw in this investigation. The obtained data are put in the wrong context. The authors compare those estuarine waters to the waters of the open ocean – why? I am told in the text that the coastal and estuarine waters are underexplored - this is hard to believe, but also easy to check. There is a lot of data on microbial communities by different approaches in coastal areas (including shotgun sequencing). Why was only the Takara Ocean data taken into consideration? It would benefit the analysis if habitats similar to the one investigated would be added. The habitat that is the Comau Fjord is poorly described in this paper. Only later I found out that there are active salmon farming facilities in the vicinity. Also some additional info would be helpful: is there a river that feeds into the fiord? what is its origin? how deep is the estuary? etc. etc. Furthermore, the sampling regime is almost nonexistent: 4 liters of water was taken at three sides - that's it! When where are the samples taken? How was it accomplished? from a ship? what equipment was used? Then there was the whole water chemistry analysis. Why was it done if it adds nothing to the paper? The data were presented in supplementary Table 1 and this is cited in line 229 - is it justified? Were those physico-chemical data taken into the PCA analysis? The title bothers me deeply. Can you make it more specific? This “unique human impacted ecosystem in Patagonia” could mean anything. At least change it to:  “A first insight into the microbial and viral communities of Comau Fjord - a unique human-impacted ecosystem in Patagonia (42 ºS)”. Throughout the text you keep repeating that these are “communities from Patagonia” - do not generalize your three samples like this. These are the surface water communities of Comau Fjord (taken at a specific time during the year). The conclusion section is absent. There is a paragraph at the end of discussion that tries to sum up the findings but the final words from the authors are vague. Is the main conclusion that there is not much data on the subject and further investigations needed to be done? Not very informative. There was a hypothesis placed in the introduction of this paper. The resolution if it holds true after the presentation and discussion of the findings should be in the conclusion section - that's the whole point. The paper needs to be rewritten and the data put into proper context with emphasis on comparison with other fjord/estuarine habitats impacted by fish farming facilities.

Author Response

Reviewer 1

The paper by Guajardo-Leiva regards the anthropogenic impact on the bacterial and viral particle diversity and function in a Patagonian estuary.

We thank the reviewer for her/his time & effort reviewing this manuscript.

At first glance the paper seems well written and composed. However even if the methodological approach is sound there is a major flaw in this investigation. The obtained data are put in the wrong context. The authors compare those estuarine waters to the waters of the open ocean – why? I am told in the text that the coastal and estuarine waters are underexplored - this is hard to believe, but also easy to check. There is a lot of data on microbial communities by different approaches in coastal areas (including shotgun sequencing). Why was only the Takara Ocean data taken into consideration?

We agree with the reviewer’s comment and apologize for not being clear. We compare samples from estuarine waters to global samples from the Tara Oceans project because we wanted to see whether there’s taxonomic and functional diversity to be uncovered by studying sites that are less represented in the scientific literature (water column samples from estuaries and fjords such as those in Patagonia). And added interest in the site is the presence of salmon farming, which is noted in the second sentence of the Abstract. To conduct such comparison, methods must be comparable, and we followed the Tara Oceans filtering and enrichment steps to make our data comparable and usable by other researchers. We now make this more explicit in the manuscript, as well as tone down the narrative that coastal and estuarine sites are largely understudied.

We acknowledge that there is some information regarding shotgun metagenomes from estuarine waters, however, most of these datasets are from waters influenced by large urban centers, and some with a long history of pollution, e.g., Delaware Bay, Chesapeake Bay, coastal China, etc. (cited in the manuscript). Most of the available literature come from sediments not water, including in Chile (https://doi.org/10.1007/s13213-017-1317-8). As pointed out above, a metanalysis of estuarine waters requires some degree of uniformity in the methods of collection and enrichment of samples, the sequencing approaches, and bioinformatic pipelines, and published studies are disparate in the methods they use. We affirm the scope of this study in the Abstract and last paragraph of the Introduction section to reflect a comparison of potential untapped taxonomic and functional diversity in the Comau Fjord and global surveys.

In addition, when analyzing the resistome, we compare our samples to surface microbial communities from the Southern Ocean, the south Atlantic and south Pacific Oceans because those are the most similar ones regarding beta diversity/species turnover according to our results showed in Figure 1. We now mention this more clearly in the manuscript.

It would benefit the analysis if habitats similar to the one investigated would be added. The habitat that is the Comau Fjord is poorly described in this paper. Only later I found out that there are active salmon farming facilities in the vicinity. Also some additional info would be helpful: is there a river that feeds into the fiord? what is its origin? how deep is the estuary? etc. etc. Furthermore, the sampling regime is almost nonexistent: 4 liters of water was taken at three sides - that's it! When where are the samples taken? How was it accomplished? from a ship? what equipment was used?

We acknowledge that our study site is only partially described in the first paragraph of the Materials and Methods section, as well as the way samples were taken. We have now added characteristics that will allow the readers better understand the site. Also, we added a more detailed description of how the samples were taken. The paragraph now reads:

“We collected 40 L of water at 5 m below sea level at 3 sampling sites (each 3 km apart; from a boat) at the head of the Comau fjord located in Northern Patagonia (42°S) using an electric water pump (Pedrollo, Italy). The fjord is characterized by seasonal variability associated with freshwater inputs from nearby rivers (Blanco, Cisnes, Lloncochaigue, Vodudahue rivers), a mean depth of 250 m (maximum of 480 m), 41 km of length and 4.5 km of width, precipitation (> 5,000 mm), and ice thawing in the summer season. During water collection in February 2016, we used a 50 mm mesh to avoid macroscopic debris. Water samples were immediately stored at 4°C for no more than 48 h until processing. Then, we assembled a filtration system using a peristaltic pump to run the 40 L of water of each sample through a 0.22 μm Sterivex filter to capture prokaryotic organisms, and subsequently through a 100 kDa tangential concentrator to concentrate viral particles. These resulted in prokaryotic and viral fractions for each of the 3 water samples, named: Bac S1/S2/S3 and Vir S1/S2/S3, respectively.”

In the second sentence of the Abstract and in the Introduction section we had pointed out that there is salmon farming in the area.

Then there was the whole water chemistry analysis. Why was it done if it adds nothing to the paper? The data were presented in supplementary Table 1 and this is cited in line 229 - is it justified? Were those physico-chemical data taken into the PCA analysis?

The water chemistry analysis was done to characterize the samples and to inform future researchers, including our own future research. Then, we decided to include them for reference to readers. These data were not included in the PCA analysis.

The title bothers me deeply. Can you make it more specific? This “unique human impacted ecosystem in Patagonia” could mean anything. At least change it to:  “A first insight into the microbial and viral communities of Comau Fjord - a unique human-impacted ecosystem in Patagonia (42 ºS)”.

We thank the reviewer for this suggestion. We have changed the title accordingly.

Throughout the text you keep repeating that these are “communities from Patagonia” - do not generalize your three samples like this. These are the surface water communities of Comau Fjord (taken at a specific time during the year).

We agree with the reviewer comment and now have changed the reference “communities from Patagonia” to “communities from the Comau Fjord”. Also, we provide more information regarding date of sampling and other supporting information (which was already present in the metadata of the bioproject but not in the manuscript).

The conclusion section is absent. There is a paragraph at the end of discussion that tries to sum up the findings but the final words from the authors are vague. Is the main conclusion that there is not much data on the subject and further investigations needed to be done? Not very informative. There was a hypothesis placed in the introduction of this paper. The resolution if it holds true after the presentation and discussion of the findings should be in the conclusion section - that's the whole point. The paper needs to be rewritten and the data put into proper context with emphasis on comparison with other fjord/estuarine habitats impacted by fish farming facilities.

Thank you for pointing this out. We have rewritten the last paragraph of the Discussion section to better reflect the conclusions of this study. It now reads:

“In this study, we tested whether Comau microbial and viral communities would be distinct in composition and function in comparison to those collected in global surveys, however, we expected they would bear the hallmarks of microbes from coastal and temperate regions. Our results support this hypothesis, as Comau Fjord microbial and viral communities are diverse and different from those sampled in global ocean surveys. Likewise, most published global ocean surveys are densely sampled in regions other than cold water and high latitude oceans, which would partially explain the novelty of the Comau Fjord and locations alike. We also hypothesized that Comau microbial communities will be functionally enriched in ARGs in general and in those related to salmon farming. This was partially confirmed by our data since we were able to find genetic determinants of resistance against florfenicol and oxytetracycline (most widely used antibiotics in salmon farming in Chile). However, our study presents limitations. More dense sampling, with a spatial and temporal design might be able to better capture any signal of disturbance by salmon farming. Also, including microbes attached to particles, as well as sediment samples might yield a higher resolution picture of the anthropogenic disturbance caused by salmon farming. Finally, a metanalysis of estuarine waters, fjords, and channels, with comparable methodologies, might render further insight as to the effect of anthropogenic disturbances in the microbial and viral communities of the water column.

Since most of human activities occur in coastal areas, such as bays, sounds, fjords, and estuaries, they represent a unique ecosystem that needs to be further studied to under-stand the interplay between human and environmental health. In particular, deter-mining whether microbial communities are resistant or resilient to disturbances of anthropogenic origin, i.e., diesel contamination, antibiotics, will allow us to understand to what extent their functions will also be affected in the future.”

Reviewer 2 Report

After reading the work of Guajardo-Leiva et al. carefully, I believe it is of considerable scientific value, performed by state-of-the-art techniques and bioinformatic analyses, and most importantly, it will be of big interest to the readers of “Microorganisms”.
However, I have some remarks which, in my opinion, should be addressed:

1. It is not mentioned if the quality of the DNA for next-generation sequencing was monitored for fragmentation. I don’t believe that pictures of electrophoresis should be added. However, such monitoring should be added within the text or be explained why it is not performed;

2. The paragraph in lines 218-222 does not contain results, so it should not be present. It is also redundant with part of the Introduction section;

3. The authors should find a way to put Figure 3 on one page;

4. I believe that the suppl. files should also be included within the article because they contain important information.

Therefore I recommend the article for publication after the minor revisions I suggest are made.

Author Response

Reviewer 2

After reading the work of Guajardo-Leiva et al. carefully, I believe it is of considerable scientific value, performed by state-of-the-art techniques and bioinformatic analyses, and most importantly, it will be of big interest to the readers of “Microorganisms”. 

We thank the reviewer for her/his time & effort and comments.

However, I have some remarks which, in my opinion, should be addressed:

  1. It is not mentioned if the quality of the DNA for next-generation sequencing was monitored for fragmentation. I don’t believe that pictures of electrophoresis should be added. However, such monitoring should be added within the text or be explained why it is not performed;

We have now added that we checked DNA integrity by capillary electrophoresis using a fragment analyzer in the Materials and Methods section.

  1. The paragraph in lines 218-222 does not contain results, so it should not be present. It is also redundant with part of the Introduction section;

Thank you for pointing this out. The paragraph has been removed.

  1. The authors should find a way to put Figure 3 on one page;

Upon acceptance, we will request the typesetting and formatting team to consider fitting the figure in one page.

  1. I believe that the suppl. files should also be included within the article because they contain important information.

We have moved the supplementary tables to the main text of the manuscript.

Therefore I recommend the article for publication after the minor revisions I suggest are made.

Round 2

Reviewer 1 Report

I accept the changes made by the authors. In my opinion the paper is sufficient for publication.